# Sensitivity to social norm violation is related to political orientation

**Élise Désilets**[1], **Benoit Brisson**[1]*, **Sébastien Hétu**[2]*

**1** Psychology department, Université du Québec à Trois-Rivières, Trois-Rivières, Québec, Canada,
**2** Psychology department, Université de Montréal, Montréal, Québec, Canada

\* Benoit.Brisson@uqtr.ca (BB); sebastien.hetu@umontreal.ca (SH)

## Abstract

Human behavior is framed by several social structures. In the present study, we focus on two of the most important determinants of social structures: social norms and political orientation. Social norms are implicit models of shared expectations about how people should behave in different social contexts. Although humans are very sensitive to violations in social norms, there are important individual differences in our sensitivity to these violations. The second concept this study focuses on is political orientation that is define by a continuum from left (liberal) to right (conservative). Individual political orientation has been found to be related to various individual traits, such as cognitive style or sensitivity to negative stimuli. Here, we propose to study the relation between sensitivity to social norm violation and political orientation. Participants completed a task presenting scenarios with different degrees of social norm violation and a questionnaire to measure their political opinions on economic and identity issues. Using hierarchical regressions, we show that individual differences in sensitivity to social norm violation are partly explained by political orientation, and more precisely by the identity axis. The more individuals have right-oriented political opinions, the more they are sensitive to social norm violation, even when multiple demographics variables are considered. Our results suggest that political orientation, especially according to identity issues, is a significant factor of individual differences in social norm processing.

## Introduction

Social structure guides human behavior through explicit and implicit rules that frame our decisions and our actions. One important determinant that contributes to social structure by facilitating large-scale coordination is social norms [1]—unwritten rules about how people should behave in society. Social norms thus structure our social environment by guiding our own behavior and by framing our expectations of how others should behave in specific situations [2]. On one hand, when social norms are well understood and respected, everyone can more easily predict the course of social interactions, which reduces the uncertainty in our social environment [3]. On the other hand, when social norms are absent or violated, it can increase this uncertainty [1]. The efficiency of social norms as a regulating mechanism is highly dependent on their enforcement: violators will often be punished [4]. Indeed, individuals are

**Data Availability Statement:** All relevant data are within the manuscript and its Supporting Information files.

**Funding:** This research was supported by a grant from the Social Science and Humanities Research Council (SSHRC; SH) and a grant from the Québec

Bio-Imaging Network (QBIN-FRQS; SH-BB). The funders had no role in study design, data collection and analysis, decision to publish, or preparation of the manuscript.

**Competing interests:** The authors have declared that no competing interests exist.

generally sensitive to social norm violation—most people are willing to punish social norm transgressors even if it comes at a cost to them [5–8]—and people who contemplate transgressing these norms are aware of these potential retaliations. However, sensitivity to social norm violation can vary across cultures and societies [9, 10] where some have stricter norms (low tolerance to social norm violation) while others may have weaker norms (stronger tolerance to norm violation) [11]. For example, variations in exposition to societal threats is associated to variation in the strength of social norms at the culture level [9]. Similar to inter-cultural differences, there are also individual differences in sensitivity to social norm violation. For example, previous work have shown that one's sensitivity is affected by the way one perceives the society he/she lives in. Indeed, individuals who think that within their society the range of permissible behavior is narrow versus wide will tend to be more sensitive to social norm violation [10, 12] suggesting that certain characteristics of how individuals perceive their social structure can be related to how they process social norms. Here we propose to investigate the possible relation between individual differences in sensitivity to social norm violation and another key concept related to social structure: political orientation.

There are several ways to conceptualize political orientation, a psychological concept that fosters affinity between individuals and surrounding political discourses and ideologies. Jost and al. (2009) argued that despite the complexity of political views, often a single left (liberal) —right (conservative) continuum, based on two interrelated aspects—advocating versus resisting social change, and rejecting versus accepting inequality—can be a sufficient, useful approximation (see also Graham, Haidt & Nosek, 2009 [13]).

Differences in political orientation has been linked to various individual traits, such as personality [14], moral foundations [13] and cognitive style [15]. According to the influential Motivated Social Cognition theory [15], these individual differences are rooted in three basic psychological needs and motivations, which are managed differently according to political orientation: (1) existential needs that encompass security, safety and comfort, (2) relational needs such as personal or social identification, interpersonal relationships and solidarity with others, and (3) epistemic needs, which refer to cognitive preference for predictability, certainty and cognitive closure (i.e., the desire to eliminate ambiguity) [16]. With the main ideas of preserving status quo and respecting/emphasising social hierarchy, a more right-oriented political orientation has been argued to serve all these needs more directly [17]. For example, in line with the epistemic needs to reduce uncertainty and ambiguity, the more individuals are right-oriented, the more they tend to have a stricter cognitive style, which may help them attain a better sense of cognitive closure [18]. Thus, to fill the need for structure and order (concepts that are related to the basic epistemic needs), right-oriented individuals will prefer conservative ideology, which favors a more rigid social structure that tends to reduce uncertainty and ambiguity through stronger regulations of our behaviors.

In sum, social norms, as well as political ideology, can be perceived as means to guide and structure social behaviour, and both can play an important role in reducing uncertainty and ambiguity. This raises the possibility that how people react to violations in social norms can be related to individual differences in political orientation. The main objective of this study was to assess the relationship between sensitivity to social norm violation and political orientation, while controlling for potential confounding socio-demographic variables (see Analyses for further details). Previous work have looked at links between political orientation and variables such as disgust [19], and sensitivity to negative non-social stimuli [20] and there are theories that suggest that ingroup loyalty, which is a important factor of compliance to social norms [21, 22], is also linked to political orientation [23]. However, there is no previous work to our knowledge that specifically studied the direct link between sensitivity to social norm violation and political orientation. We hypothesised that the more an individual is right-oriented, the

more he or she will be sensitive to social norm violations, given that sensitivity helps detect violators and reinforce social norms which in turn favors a more rigid social structure.

Although political orientation has been often tapped on a single left-right continuum, it has been argued that conceptualizing political orientation on more dimensions could lead to a more nuanced understanding of its link to psychological factors [24–26]. Even when political orientation is used as a single continuum, two core aspects are used to describe the differences between left and right: preference for social change versus tradition and preference for equality versus inequality [16, 27]. These two aspects can be mirrored respectively to social and economic dimensions [25], which have been argued to be linked differentially to a wide range of variables [21]. In this regard, we have explored our main question by using a political questionnaire that was conceived to measure identity issues (social dimension) and socioeconomic issues (economic dimension) separately into two axes [28]. The decision to focusing on identity and socioeconomic issues to distinguish between the social and economic dimensions of political orientation was primarily based on the fact that the present study was conducted in the province of Quebec (Canada). First, identity issues have been at the core of Quebec politics throughout its history, and were instrumental for French-Canadians of European descent (historically subordinate in Canada) to acquire political agency in a province where they have always constituted the demographic majority [29]. Second, left-wing socio-democratic economic policies have colored the political environment of the province at least since the Quiet Revolution (*Révolution tranquille*, 1960–70). For these two reasons, we expected that the chosen axes could be dissociated within our sample—Quebec participants potentially being more left-wing in the socioeconomic axis than in the identity axis. Given that social norms have a significant affective component (violations producing aversive feelings [30]), we predicted that if the two axes could be dissociated, sensitivity to social norm violation would be linked more strongly to the identity axis than the socioeconomic axis, based on the notion that identity issues are more emotionally salient than socioeconomic issues [24, 25].

## Methods

### Participants

One hundred and ninety-nine French-speaking participants (French being the first language of 196 participants; 145 identifying as women and 54 identifying as men based on a two-alternative forced choice question) aged between 19–54 years (M = 24.33 years, SD = 6.68 years) completed this study. This sample is composed from three recruitment phases that took place between winter 2018 and fall 2019. Participants were recruited at the Université du Québec à Trois-Rivières and the Université de Montréal in the province of Quebec, Canada. Further descriptive statistics about the participants are presented in Table 1. This study was approved by the Comité d'éthique à la recherche avec des êtres humains (CEREH; Université du Québec à Trois-Rivières) and by the Comité d'éthique de la recherche en éducation et en psychologie (CEREP; Université de Montréal). Before participating, all participants gave their written consent. Participants received a monetary compensation for their participation.

**Table 1. Descriptive statistics.**

| Place of birth | | Education level (years) | | | | Occupation | | | Living environment | |
|---|---|---|---|---|---|---|---|---|---|---|
| Quebec | Other | min | max | mean | sd | Student | Employed | Other | Urban | Rural |
| 177 | 22 | 11 | 30 | 16.82 | 2.91 | 175 | 21 | 3 | 152 | 47 |

## Stimuli and procedure

Participants had to complete a Social Norm Violation task [10] aimed at measuring individual sensitivity to social norm violation on a computer and answer socio-demographic and political questionnaires. Note that participants also completed another task aimed at measuring social decision-making and costly punishment (Ultimatum game [31]) but that only results regarding the Social Norm Violation task are described here. Questionnaires and tasks were presented on a computer using E-prime 2.0 (Sharpsburg, Psychology Software Tools, Inc.). The order in which participants had to complete the questionnaires and tasks was counterbalanced.

**Social norm violation task.** Participants completed a Social Norm Violation task developed by Mu et al. (2015) where they had to judge real-life scenarios. Scenarios from the original task were translated from English to French. A total of 34 behaviors (e.g., yelling) were presented in three situations/conditions: appropriate (e.g., at a rock concert), weakly inappropriate (e.g., on the metro) and strongly inappropriate (e.g., in the library) creating 102 scenarios (e.g., Steven is at a rock concert. He is yelling.). The translated original scenarios were first tested in pilot studies with French-speaking university students and six where adjusted to better reflect reality in the province of Quebec. Examples of the scenarios are presented in Table 2. Scenarios were randomly presented to minimize the risk of participants discovering the structure of the experiment (e.g., that each behavior was paired with three different situations). On each trial, participants had to rate the appropriateness of each scenario on a four point scale ranging from 1 (strongly appropriate) to 4 (strongly inappropriate). The task's timeline is presented in Fig 1. As proposed by Mu et al. (2015), individual sensitivity social norm violation was measured by calculating the percentage of strongly inappropriate scenarios that were rated by the participant as strongly inappropriate.

**Political orientation.** To assess individual political orientation, participants completed a modified version of the Cald questionnaire [28] (see also [32]). This modified version is meant to assess political opinions using 8 items separated into two political axes: a) socioeconomic (4 items: taxes; globalization; public services and role of the state and business), and b) identity (4 items: right to vote and nationality; fight against delinquency; poverty and foreclosure; and immigration). Each item was presented with different policies associated with a score from 1 (left) to 5(right) (e.g., right to vote and nationality item: All foreigners residing in Canada must have the right to vote, regardless of their nationality (1); Only Canadians must have the right to vote; and, with exceptions, we cannot be Canadian without having Canadian parents: we must apply the "blood right" and not "territorial right" (5)). For each item, participants selected the policy they most agreed with. The global score for this questionnaire was obtained

**Table 2. Scenarios example.** 5 behaviors with their 3 associate conditions and the French translation.

| Appropriate | Strongly inappropriate | Weakly inappropriate | Behavior |
|---|---|---|---|
| Jacob in the bike lane.*Jacob est sur la piste cyclable* | Jacob is on the Highway. *Jacob est sur l'autoroute* | Jacob is on the city sidewalk. *Jacob est sur le trottoir* | Cycling *Il fait du vélo* |
| Amanda is at a Tango lesson. *Amanda est à un cours de Tango* | Amanda is at the art museum. *Amanda est au musée d'art* | Amanda is on a subway platform. *Amanda est sur la plateforme du métro* | Dancing *Elle danse* |
| Thomas is at a bar. *Thomas est dans un bar.* | Thomas is at the doctor's office. *Thomas est au bureau du docteur.* | Thomas is at the post office. *Thomas est au bureau de poste.* | Flirting *Il flirte* |
| Emma is at a wedding. *Emma est à un mariage.* | Emma is at the Doctor's office. *Emma est dans le bureau du docteur.* | Emma is on a bus. *Emma est dans l'autobus.* | Kissing *Elle embrasse* |
| Chris is at the church. *Christian est à l'église.* | Chris is at the symphony. *Christian est à un concert symphonique.* | Chris is in the public park. *Christian est dans un parc public.* | Praying *Il prie* |

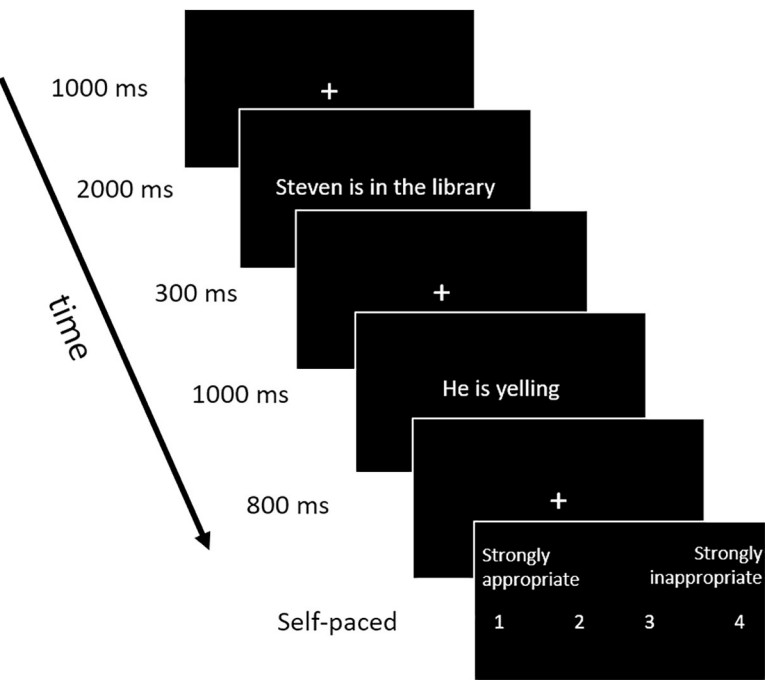

**Fig 1. Example of a trial in the social norm violation task's.**

by calculating the mean of all eight items resulting in a score from 1 (left) to 5 (right). The scores for each axis was calculated by computing the mean of the four items associated to each axis independently, also resulting in a score from 1 (left) to 5 (right) for each axis.

## Analysis

One scenario from the strongly inappropriate condition was removed from the analyses given that no participant rated it as strongly inappropriate. Note that this decision did not affect our results and conclusions. In order to investigate the possible relation between global political orientation measures and sensitivity to social norm violation, we ran a hierarchical regression analysis ($\alpha = 0.05$) —where the predicted variable was the sensitivity to norm violation—controlling for the possible influence of sociodemographic (education level and gender) and procedure-related (task/questionnaire order of completion) variables. These variables were entered in the first stage of the analysis and global political orientation was entered in the second stage. As described below, in a second step, partial correlations revealed that only the identity axis–and not the socioeconomic axis—was related to sensitivity to social norm violation. A second hierarchical regression was therefore conducted to assess the relation between the identity axis and sensitivity to social norm violation while controlling for possible influences of sociodemographic, procedure-related variables as well as socioeconomic political orientation.

## Results

### Political orientation and sensitivity to social norm violation

The group's global political orientation scores, as well as for identity and socioeconomic axes independently, in addition to the mean percentage of scenarios rated as strongly inappropriate

**Table 3. Descriptive statistics.**

| | Mean | Median | SD | Min | Max |
|---|---|---|---|---|---|
| General political orientation | 2.34 | 2.38 | 0.61 | 1.00 | 3.75 |
| Socioeconomic axis | 2.25 | 2.25 | 0.80 | 1.00 | 4.25 |
| Identity axis | 2.52 | 2.50 | 0.69 | 1.00 | 4.25 |
| Percentage of scenarios rated as strongly inappropriate | 40.69 | 39.39 | 18.63 | 0.00 | 84.85 |

are presented in Table 3. Our sample's mean global political orientation scores are more on the left of the political centre (3.0) (p < .001) which was predictable considering that our sample was in vast majority composed of Quebec university students which have been found to be more left-oriented (see [32] for similar results).

## Relation between global political orientation and sensitivity to social norm violation

A significant positive correlation was found between global political orientation scores and sensitivity to social norm violation, with right political orientation linked to greater sensitivity (Table 4).

In order to isolate the contribution of global political orientation on scores of sensitivity to social norm violation, a hierarchical regression was performed. The general model, including demographics variables (education level and gender), testing-related variables (order of completion) and our variable of interest (global political orientation), was significant ($F_{(4,194)}$ = 7.11, $p < 0.001$) and explained 11.00% of the variance in sensitivity to social norm violation. Global political orientation contributed significantly to the model ($\Delta F_{(1,194)}$ = 9.50, $p = 0.002$) and accounted for 4.30% of the variance in the model. Hierarchical regression results are presented in Table 5.

## Relation between identity axis and sensitivity to social norm violation

As seen in Table 3, identity and socioeconomic axes were found to be positively correlated and share 15% of variance. Furthermore, as hypothesised, participants in our sample were more left-oriented on the socioeconomic axis than on the identity axis ($t_{(198)}$ = 4.52, $p < 0.001$).

Hence, partial correlations were conducted for each axis in order to control for covariance of the other axis when exploring the link between each axis and sensitivity to social norm violation. These analyses revealed a significant correlation between the identity axis and sensitivity to norm violation ($r_{(197)}$ = 0.14, $p = 0.04$), but not for the socioeconomic axis ($r_{(197)}$ = 0.09, $p = 0.23$).

**Table 4. Correlations between political orientation scores and sensitivity to social norm violations.**

| | Percentage of scenarios rated as strongly inappropriate | General political orientation | Socioeconomic axis | Identity axis |
|---|---|---|---|---|
| Percentage of scenarios rated as strongly inappropriate | — | 0.18* | 0.15* | 0.19** |
| General political orientation | | — | 0.78*** | 0.85*** |
| Socioeconomic axis | | | — | 0.39*** |
| Identity axis | | | | — |

*p < 0,05

**p < 0.01

***p < 0.001.

**Table 5. Coefficients of hierarchical regression analysis for variables predicting sensitivity to social norm violation with general political orientation (n = 199).**

| Model | | Unstandardized β | Standard Error | Standardized β | T | 95% CI Lower | 95% CI Upper |
|---|---|---|---|---|---|---|---|
| H₀ | (Intercept) | 39.48 | 8.59 | | 4.60*** | 22.55 | 56.42 |
| | Gender | 1.58 | 0.72 | 0.15 | 2.20* | 0.17 | 3.00 |
| | Education level | -0.84 | 0.44 | -0.13 | -1.92 | -1.71 | 0.02 |
| | Order | 3.66 | 1.12 | 0.22 | 3.27** | 1.45 | 5.86 |
| H₁ | (Intercept) | 25.62 | 9.53 | | 2.69** | 6.81 | 44.42 |
| | Gender | 1.75 | 0.71 | 0.17 | 2.48* | 0.35 | 3.14 |
| | Education level | -0.95 | 0.43 | -0.15 | -2.21* | -1.81 | -0.10 |
| | Order | 3.73 | 1.09 | 0.23 | 3.41*** | 1.57 | 5.89 |
| | Global political orientation | 6.38 | 2.07 | 0.21 | 3.08** | 2.30 | 10.46 |

*p < 0,05

**p < 0.01

***p < 0.001.

Multicollinearity was respected as indicated by a variance inflation factors (VIFs) all near to the recommended value of 1 (min: 1.00, max: 1.01). Autocorrelation in the residuals was also respected as expressed by a value of 1.95 for the Durbin-Watson test. The residuals' mean is 0 and residuals present no problematic value (all value has a cook's distance <1).

Since the identity axis was linked with sensitivity to social norm violation, we conducted a second hierarchical regression using identity scores instead of global political orientation scores. For this second regression, socioeconomic scores were added in the first step of the regression with the demographics variables previously mentioned. The general model was significant (F (5,193) = 6.38, p < 0.001) explaining 12.0% of the variance in sensitivity to social norm violation. The Identity axis contributed significantly to the model (ΔF(1,193) = 4.08, p = 0.045) and explained 1.8% of the variance. Hierarchical regression results are presented in Table 6.

**Table 6. Coefficients of hierarchical regression analysis for variables predicting sensitivity to social norm violation with identity and socioeconomic axes (n = 199).**

| Model | | Unstandardized β | Standard Error | Standardized β | T | 95% CI Lower | 95% CI Upper |
|---|---|---|---|---|---|---|---|
| H₀ | (Intercept) | 30.27 | 9.00 | | 3.37*** | 12.53 | 48.01 |
| | Gender | 1.77 | 0.71 | 0.17 | 2.50* | 0.37 | 3.16 |
| | Education level | -0.99 | 0.43 | -0.15 | -2.28* | -1.85 | -0.13 |
| | Order | 3.89 | 1.10 | 0.24 | 3.54*** | 1.72 | 6.06 |
| | Socioeconomic axis | 4.62 | 1.58 | 0.20 | 2.92** | 1.50 | 7.74 |
| H₁ | (Intercept) | 23.32 | 9.57 | | 2.44* | 4.45 | 42.19 |
| | Gender | 1.82 | 0.70 | 0.17 | 2.59* | 0.43 | 3.21 |
| | Education level | -0.98 | 0.43 | -0.15 | -2.29* | -1.83 | -0.14 |
| | Order | 3.78 | 1.09 | 0.23 | 3.46*** | 1.62 | 5.94 |
| | Socioeconomic axis | 3.29 | 1.70 | 0.14 | 1.93 | -0.07 | 6.65 |
| | Identity axis | 3.94 | 1.95 | 0.15 | 2.01* | 0.09 | 7.79 |

*p < 0,05

**p < 0.01

***p < 0.001.

Multicollinearity was respected as indicated by a variance inflation factors (VIFs) all near to the recommended value of 1 (min: 1.01, max: 1.21). Autocorrelation in the residuals was also respected as expressed by a value of 1.89 for the Durbin-Watson test. The residuals' mean is 0 and residuals present no problematic value (all value has a cook's distance <1).

## Discussion

The main goal of the present study was to explore the link between political orientation and sensitivity to social norm violation. Since both right-wing political ideology and sensitivity to social norm violation can be powerful tools to reduce ambiguity and uncertainty in one's social environment [16, 17, 33, 34], we hypothesised that right political orientation would be positively associated to sensitivity to social norm violation. This hypothesis was confirmed in the present study, even when controlling for education level, and gender.

These results extend to the social domain a large body of evidence that linked political orientation with non-social information processing [17]. Indeed, individual political orientation has been found to be related to specific cognitive processes such as conflict-related anterior cingulate activity [35], perceptual bias [32, 35] and to different cognitive styles [15]. In the current study, participants had to process relatively complex social information contained in real-life scenarios presenting others behaving in various contexts. This type of processing is generally more complex than the tasks or questionnaires used in previous studies on political orientation and information processing. To our knowledge, this is the first study to link political orientation to the processing of social norms.

The present results are also in line with previous studies that have suggested that the more individuals are on the right of the political spectrum, the more they are sensitive to a range of negative stimuli, such as angry faces, negative words or images [20]. Our results, linking right political orientation and sensitivity to norm violation measured using real-life social scenarios, suggest that this relation can be extended to relatively complex negative stimuli related to social interactions and behaviors.

Overall, results from the present study support the Social Motivated Cognitive theory [15, 17], as higher sensitivity to social norm violation could help fulfill the three basic needs and motivations which are proposed to be served more directly by right-wing ideology. First, regarding relational needs, detecting violations more efficiently could help identify individuals that adhere and respect the same internalized norms, which in turn could reinforce the feeling of shared reality and belonging with these individuals [36]. Second, for existential motivations, greater sensitivity to social norm violation could also help fulfill the need for safety and security by giving an impression of control on the environment, in that the better one is at detecting social norm violations, the better one is suited to react to rectify the situation [37]. Finally, in respect to epistemic motivations, greater sensitivity to social norm violation could help reduce uncertainty and ambiguity by reducing the range of possible behaviors one can encounter and helping one choose how to act in the proper dictated manner [3, 34].

The present results also fit within the Moral Foundation theory of political identity [13], which has proposed that right-wing individuals' moral foundations encompass all five moral intuitions, including Ingroup/loyalty, Authority/respect, Purity/sanctity, Harm/care and Fairness/reciprocity, whereas left-oriented individuals endorse and use the two later more. The three morals foundations used more strongly by the right-oriented individuals (Ingroup/loyalty, Authority/respect and Purity/sanctity) emphasise group binding, loyalty and self-control. The importance given to these foundations might influence or be influenced by decision-making and actions related to social norms. For instance, giving higher importance to in-group binding could bring one to adopt stricter social norms and greater sensitivity could help detect more easily violators—individuals who are potentially not part of one's own group [33]. In addition, research has shown that social norms can influence loyalty behavior in virtual community where the more people believe in the importance of community loyalty behavior, the more they individually act in accordance with the group's norms [38]. Finally, individuals giving higher importance to self-control as a desired characteristic for members of their society

could be more sensitive to social norm violations as violations could indicate a lack of self-control in others. Indeed, less self-control has been linked with higher social norms violation [39, 40]. Interestingly, it is possible that participants used the three binding foundations to evaluate the appropriateness of the scenarios presented in the Social Norm Violation task. Indeed, some scenarios were linked with ingroup behaviors (ex. Situations with family), respect for authority (ex. Situations with boss) and with purity (ex. Situations including swearing). If this is the case, since right oriented individuals use these foundations more, they could judge violations in these contexts as more inappropriate. However, since we did not measure the Moral Foundations of our participants or directly ask them how they made their decisions, this hypothesis remains to be tested.

A second objective of the present study was to explore whether socioeconomic (where political opinions are grounded on respect to economic structure) and identity (where political opinions are based more on respect to social values [24, 25]) political opinions could be dissociated in our sample mostly composed of individuals born in the province of Quebec. If so, we also wanted to investigate whether sensitivity to social norm violation would be linked more strongly to the identity axis than to the socioeconomic axis. As hypothesised, the two axes were moderately correlated and the mean of the socioeconomic axis scores was slightly but significantly more to the left than scores on the identity axis. More importantly, only the identity axis was linked to sensitivity in social norm violation when controlling for the socioeconomic axis with a partial correlation. Which supports the assumption that social norms, as measured in the present study, are more related to in-group identity and loyalty issues than socioeconomic issues. However, although the results from the hierarchical regressions also support a greater role of the identity axis, compared to the socioeconomic axis in the relation between political orientation and sensitivity to social norm violation, observed standardized betas, a measure of effect size, for the two axes in this analysis are quite similar. This indicates that sociodemographic variables may be important to consider in further studies to better understand the differential link of each political axis, as observed in this study.

It is important to mention that our sample was mainly composed of university students of French-Canadian lineage. These two demographics particularities make the results difficult to generalize outside the educated French-Canadian majority population of the province of Quebec. However, the relative homogeneity of our sample allowed us to explore possible differences between different dimensions of political orientation. Indeed, whereas some propose that political orientation can be conceptualized as a single left-right continuum [13, 16], we found that within our educated French-speaking Quebec sample, political orientation related to socioeconomic and identity issues could be distinguished, allowing us to describe more precisely how social norms sensitivity was related to the identity axis of political orientation while not with the socioeconomic axis in the present study. Such results add to mounting evidence that social and economic dimensions of political orientation are linked differently to multiple variables [21].

This research opens different possibilities in the field of social norms processing by showing that sensitivity to social norm violation can be linked to political orientation and possibly more to the identity axis. Given that our sample focused on a relatively homogeneous group, it will be important to examine whether these findings can be observed in different socio-demographic groups and political realities. Also, as this research used a correlational design, no causality can be inferred, and further research will be required to investigate the direction of the observed relation between social norm violation and political orientation when observed. Results presented here focussed on general sensitivity to social norm violation. However, it would also be interesting to study if different types of social norm violations have stronger relationships with political orientation than others.

## Supporting information

**S1 File Data. All data underlying the findings.**
(CSV)

**S2 File Questionnaire. Political orientation questionnaire.**
(DOCX)

## Acknowledgments

We wish to thank Yan Mu et Michelle Gelfand for sharing the scenarios used in the Social Norm Violation task and the research assistants (Lesly-Anne Chabot and Lydia Nait-Said) who helped in collecting data.

## Author Contributions

**Conceptualization:** Benoit Brisson, Sébastien Hétu.

**Formal analysis:** Élise Désilets.

**Funding acquisition:** Sébastien Hétu.

**Investigation:** Élise Désilets.

**Methodology:** Élise Désilets.

**Project administration:** Sébastien Hétu.

**Resources:** Sébastien Hétu.

**Supervision:** Benoit Brisson, Sébastien Hétu.

**Writing – original draft:** Élise Désilets.

**Writing – review & editing:** Élise Désilets, Benoit Brisson, Sébastien Hétu.

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
