## [Decision Letter · Decision Letter 0]

3 Sep 2020

PONE-D-20-22085

Sensitivity to social norm violation is related to political orientation

PLOS ONE

Dear Dr. Désilets,

Thank you for submitting your manuscript to PLOS ONE. After careful consideration, we feel that it has merit but does not fully meet PLOS ONE’s publication criteria as it currently stands. Therefore, we invite you to submit a revised version of the manuscript that addresses the points raised during the review process.

I would like to see a revision of this manuscript. You have two reviews and both find some merit in the manuscript. In your revision I would like you to address the following concerns.

First, as suggested by R2, can you more clearly delineate the contribution of this manuscript to the literature? As that reviewer notes, there are quite a few studies that demonstrate the effect of political ideology on in and out-group members.

Second, can you further develop the effect sizes of your treatments?  The regressions (particularly Tables 4 and 5) detail the coefficients. In your discussions you are comfortable talking about statistical significance and explained variance. But less clear to me (and to R1) is what does this mean substantively? Hierarchical regressions can sometimes be difficult to decipher, so help me out here.

Third, you might consider R1’s suggestion of a second study, but I am not keen on asking for an additional study. It would be useful, however, to note why R1’s concern is not justified given the current study.

I look forward to a revision.

We look forward to receiving your revised manuscript.

Kind regards,

Rick K. Wilson, Ph.D.

Academic Editor

PLOS ONE

Journal Requirements:

Reviewers' comments:

Reviewer's Responses to Questions

**Comments to the Author**

1. Is the manuscript technically sound, and do the data support the conclusions?

Reviewer #1: No

Reviewer #2: Yes

2. Has the statistical analysis been performed appropriately and rigorously? 

Reviewer #1: I Don't Know

Reviewer #2: Yes

3. Have the authors made all data underlying the findings in their manuscript fully available?

Reviewer #1: Yes

Reviewer #2: Yes

4. Is the manuscript presented in an intelligible fashion and written in standard English?

Reviewer #1: Yes

Reviewer #2: Yes

5. Review Comments to the Author

Reviewer #1: The introduction is well written, clearly spells out the hypothesis, and the rational behind the main hypothesis.

I would like to see the results of the analyses without the 1 item removed from the social norm violation scale in a supplemental file or footnote.

I am not convinced by the data that socioeconomic conservatism is unrelated/or less related to norm violation compared to identity conservatism. In the regression that tests this, the effect of the identity axis is marginal (p = .045). The effect sizes (standardized beta) for both axes are different by only .01 (.15 vs .14). This is not enough of an effect for me to be confortable with the conclusions made in this paper based on only one study alone.

If would like to see another study attached to this paper that can replicate the identity vs. socioeconomic axes effect. I am not always against 1 off studies, but the effects here are not strong enough for me to feel comfortable concluding that the identity axis of ideology is more related to sensitivity to norm violation than the socioeconomic axis.

Reviewer #2: This paper presents a solid analysis of the question posed: is political ideology associated with social norm violation sensitivity. I have some questions as to how novel a question this is to pose. Prior work has documented a link between political foundations and disgust sensitivity (Brenner & Inbar, 2015) and between social norm violations and dehumanization and disgust (Fincher & Tetlock, 2016) so it seems straightforward to link the two. Further Moral Foundations, which the authors discuss, documents ingroup/loyalty as one of the core foundations for conservatives. It’s hard to imagine within the realms of current theory on groups and social norms, a group in which great ingroup loyalty wouldn’t predict greater outrage at norm violations to that ingroup. In addition, the general link between conservatives and traditional values suggests some element of preventing or being angered by those who violate norms (e.g. by being gay) such that it seems plausible that greater norm sensitivity could cause one to become more conservative politically. That said, though many papers seem to strongly suggest a positive association between political conservatism and social norm violation sensitivity, I know of no paper that directly documents this relationship and the authors do a good job in doing so.

I would suggest including the table of norm violation stimuli in the paper rather than in the supplement. I also think it could be quite interesting to do some exploratory coding of which violations have stronger relationships with the predictor variables than others. I think it would be good to have some discussion of causality in the general discussion, especially to discuss ways in which perceptions of norm violations may lead to changes in political orientation. The authors do a good job of not making causal claims but it would be useful to engage with this further. I’m also a bit confused about the large order effect in the paper – perhaps the authors could explain that further to help readers make sense of it.

6. PLOS authors have the option to publish the peer review history of their article (what does this mean?). If published, this will include your full peer review and any attached files.

Reviewer #1: No

Reviewer #2: No

---

## [Author Response · Author response to Decision Letter 0]

25 Sep 2020

Editorial comments

E.1 First, as suggested by R2, can you more clearly delineate the contribution of this manuscript to the literature? As that reviewer notes, there are quite a few studies that demonstrate the effect of political ideology on in and out-group members.

We added precision in the introduction of the manuscript highlighting that previous work mainly looked at indirect links between political orientation and social norm sensitivity. Although previous studies have demonstrated a link between political orientation and in/out-group members, the link between political orientation and social norm sensitivity per se has never been, to our knowledge, explored before this study. See p.4

Previous work have looked at links between political orientation and variables such as disgust [19], and sensitivity to negative non-social stimuli [20] and there are theories that suggest that ingroup loyalty, which is a important factor of compliance to social norms [21, 22], is also linked to political orientation [23]. However, there is no previous work to our knowledge that specifically studied the direct link between sensitivity to social norm violation and political orientation. 

E.2 Second, can you further develop the effect sizes of your treatments? The regressions (particularly Tables 4 and 5) detail the coefficients. In your discussions you are comfortable talking about statistical significance and explained variance. But less clear to me (and to R1) is what does this mean substantively? Hierarchical regressions can sometimes be difficult to decipher, so help me out here.

Both standardized and unstandardized coefficients (or betas) can be used as effect size for IV in a hierarchical regression. Unstandardized coefficients indicate how much the variable changes when the IV changes by 1 unit. Standardized coefficients have mainly the same goal, but they put effects on a common scale using standard deviation allowing to compare effect size between IV. There is no scale indicating if a beta is small, large or medium. We can only say if an effect is bigger than another according to the standardized beta. 

There is no clear prescription about what must be reported in a publication. Here, we chose to put coefficients in a table to present in a clearer way all coefficient, standardized and unstandardized. We do not mention explicitly the coefficients in the text because they are already presented in the table. 

Because our main hypothesis relates to the relation between political orientation and sensitivity to social norm violations (and not how this link is bigger or smaller than with other possible variables), we propose to keep our result and discussion mainly centered around % of variance explained which we think should help readers evaluate the “strength” of this relation. 

When discussing our results regarding socio-economic and identity axes of political orientation, we now mention that although only the relation between identity axis and sensitivity is significant, since the standardized betas are very close (a measure of effect size), these results have to be considered with caution (see R1.2). 

More importantly, only the identity axis was linked to sensitivity in social norm violation when controlling for the socioeconomic axis with a partial correlation. Which supports the assumption that social norms, as measured in the present study, are more related to in-group identity and loyalty issues than socioeconomic issues. However, although the results from the hierarchical regressions also support a greater role of the identity axis, compared to the socioeconomic axis in the relation between political orientation and sensitivity to social norm violation, observed standardized betas, a measure of effect size, for the two axes in this analysis are quite similar. This indicates that sociodemographic variables may be important to consider in further studies to better understand the differential link of each political axis, as observed in this study.

E.3 Third, you might consider R1’s suggestion of a second study, but I am not keen on asking for an additional study. It would be useful, however, to note why R1’s concern is not justified given the current study.

Please see answer to comment R1.3.

Reviewer #1: The introduction is well written, clearly spells out the hypothesis, and the rational behind the main hypothesis. 

Thank you

R1.1 I would like to see the results of the analyses without the 1 item removed from the social norm violation scale in a supplemental file or footnote. 

We reran all analyses with the item included. Our conclusions remain the same as the results are very similar (i.e., changes in values but not in significance vs. no significance). See the results with the item presented at the end of our responses to reviewers. Highlighted in yellow are the differences between these analyses and the ones presented in the text. We added the information that our choice to remove this problematic item did not qualitatively change our results or conclusions in the text. See p. 10

One scenario from the strongly inappropriate condition was removed from the analyses given that no participant rated it as strongly inappropriate. Note that this decision did not affect our results and conclusions.

R1.2 I am not convinced by the data that socioeconomic conservatism is unrelated/or less related to norm violation compared to identity conservatism. In the regression that tests this, the effect of the identity axis is marginal (p = .045). The effect sizes (standardized beta) for both axes are different by only .01 (.15 vs .14). This is not enough of an effect for me to be confortable with the conclusions made in this paper based on only one study alone. 

Although the relation between both axes and sensitivity to social norm violation is clearer when looking at our partial correlations results, we agree that the hierarchical regression results are less clear. We added some precision in the discussion about theses differences. See E.2 for more explanations and p.19-20-21 for modifications

More importantly, only the identity axis was linked to sensitivity in social norm violation when controlling for the socioeconomic axis with a partial correlation. Which supports the assumption that social norms, as measured in the present study, are more related to in-group identity and loyalty issues than socioeconomic issues. However, although the results from the hierarchical regressions also support a greater role of the identity axis, compared to the socioeconomic axis in the relation between political orientation and sensitivity to social norm violation, observed standardized betas, a measure of effect size, for the two axes in this analysis are quite similar. This indicates that sociodemographic variables may be important to consider in further studies to better understand the differential link of each political axis, as observed in this study.

[…]

This research opens different possibilities in the field of social norms processing by showing that sensitivity to social norm violation can be linked to political orientation and possibly more to the identity axis.

R1.3 If would like to see another study attached to this paper that can replicate the identity vs. socioeconomic axes effect. I am not always against 1 off studies, but the effects here are not strong enough for me to feel comfortable concluding that the identity axis of ideology is more related to sensitivity to norm violation than the socioeconomic axis.

Since our primary goal was to assess the link between political orientation and social norm sensitivity, we consider that our results are strong enough to support our conclusion about the main question presented in the paper. We agree that a second study looking into the specific relation between socioeconomic vs. identity axis and sensitivity to social norm violation would greatly benefit from a follow-up study. However, as mentioned in the manuscript, since this objective is only secondary, we decided not to run a replication study at this time, which would have further been complicated by the pandemic. We modified the discussion section to highlight that results about the different axes need to be considered with caution and that future work is necessary to confirm this result. See p. 20 

However, although the results from the hierarchical regressions also support a greater role of the identity axis, compared to the socioeconomic axis in the relation between political orientation and sensitivity to social norm violation, observed standardized betas, a measure of effect size, for the two axes in this analysis are quite similar. This indicates that sociodemographic variables may be important to consider in further studies to better understand the differential link of each political axis, as observed in this study.

Reviewer #2: This paper presents a solid analysis of the question posed: is political ideology associated with social norm violation sensitivity. 

R2.1 I have some questions as to how novel a question this is to pose. Prior work has documented a link between political foundations and disgust sensitivity (Brenner & Inbar, 2015) and between social norm violations and dehumanization and disgust (Fincher & Tetlock, 2016) so it seems straightforward to link the two. Further Moral Foundations, which the authors discuss, documents ingroup/loyalty as one of the core foundations for conservatives. It’s hard to imagine within the realms of current theory on groups and social norms, a group in which great ingroup loyalty wouldn’t predict greater outrage at norm violations to that ingroup. In addition, the general link between conservatives and traditional values suggests some element of preventing or being angered by those who violate norms (e.g. by being gay) such that it seems plausible that greater norm sensitivity could cause one to become more conservative politically. That said, though many papers seem to strongly suggest a positive association between political conservatism and social norm violation sensitivity, I know of no paper that directly documents this relationship and the authors do a good job in doing so. 

Some of these considerations were already mentioned in the discussion, but we agree that the novelty of this study was not put forward in the introduction of the paper. Therefore, we added some precisions in the introduction based on the references the reviewer suggested in order to better show the novelty of our study in relation to previous studies. See p.4

Previous work have looked at links between political orientation and variables such as disgust [19], and sensitivity to negative non-social stimuli [20] and there are theories that suggest that ingroup loyalty, which is a important factor of compliance to social norms [21, 22], is also linked to political orientation [23]. However, there is no previous work to our knowledge that specifically studied the direct link between sensitivity to social norm violation and political orientation. 

R2.2 I would suggest including the table of norm violation stimuli in the paper rather than in the supplement. 

As suggested, the table is now included in the manuscript. See p. 9

R2.3 I also think it could be quite interesting to do some exploratory coding of which violations have stronger relationships with the predictor variables than others. 

We agree that it would be quite interesting to explore the link between specific scenarios or “types” of scenarios and political orientation. However, our paradigm is not optimized for this. Indeed, as each scenario is rated on a scale from 1 to 4, there is very little variance in the answers within our sample which greatly hinders the use (and validity) of correlation analysis at the scenario level. However, we added this suggestion to the future studies at the end of the discussion. See p.21

Results presented here focussed on general sensitivity to social norm violation. However, it would also be interesting to study if different types of social norm violations have stronger relationships with political orientation than others. 

R2.4 I think it would be good to have some discussion of causality in the general discussion, especially to discuss ways in which perceptions of norm violations may lead to changes in political orientation. The authors do a good job of not making causal claims but it would be useful to engage with this further. 

It is difficult to assess causality when it comes to political orientation. We think that it is unclear if it is sensitivity to social norm violations or political orientation which has an influence on the other variable. Even if the causality is an important theoretical (and practical) question, the design used in this study does not allow us to measure causality evidence, as is the case with the majority of studies on political orientation. Therefore, we are not comfortable engaging in a discussion based on untested hypothesis.

This being said, Hibbing et al. (2014) and Inbar et al. (2009) argue that political orientation could be influenced by negative bias (définir en deux mots et si possible en parlant de sensibilité) more than the other way based on a longitudinal study in which there was correlations between childhood temperament and adult political beliefs and some literature suggesting that changes in the negative features of an environment or negative events such as terrorist attacks make people more conservative. Following this line of thinking, one could think that our political orientation could be influenced by our sensitivity to social norm violation but we prefer not to discuss this in the article considering that our correlational design did not allow for testing causality.

R2.5 I’m also a bit confused about the large order effect in the paper – perhaps the authors could explain that further to help readers make sense of it.

The order effect is indeed a surprising result. It seems that participants became less sensitive with time: sensitivity seemed to diminish as a function of how “late” in the experimental session the social norm violation task was completed. We did not want to emphasise this result in the manuscript because we think that it is not related to our main question. However, if you consider that this information is essential to the paper, we will add it with pleasure.

Supplementary analysis

Table 3. Correlations between political orientation scores and sensitivity to social norm violations

 Percentage of scenarios rated as strongly inappropriate General political orientation Socioeconomic axis Identity axis

Percentage of scenarios rated as strongly inappropriate --- 0.18* 0.15* 0.19**

General political orientation --- 0.78*** 0.85***

Socioeconomic axis --- 0.39***

Identity axis ---

In order to isolate the contribution of global political orientation on scores of sensitivity to social norm violation, a hierarchical regression was performed. The general model, including demographics variables (education level and gender), testing-related variables (order of completion) and our variable of interest (global political orientation), was significant (F(4,194) = 7.18, p < 0.001) and explained 11.10% of the variance in sensitivity to social norm violation. Global political orientation contributed significantly to the model (ΔF(1,194) = 9.50, p = 0.002) and accounted for 4.30% of the variance in the model. Hierarchical regression results are presented in Table 4.

Table 4. Coefficients of hierarchical regression analysis for variables predicting sensitivity to social norm violation with general political orientation (n=199)

 95% CI

Model Unstandardized β Standard Error Standardized β t Lower Upper

H₀ (Intercept) 38.21 8.33 4.59*** 21.77 54.64 

 Gender 1.55 0.70 0.15 2.22* 0.17 2.92 

 Education level -0.82 0.43 -0.13 -1.92 -1.66 0.02 

 Order 3.58 1.09 0.23 3.30** 1.44 5.73 

H₁ (Intercept) 24.74 9.25 2.67** 6.49 43.00 

 Gender 1.71 0.69 0.17 2.49* 0.36 3.06 

 Education level -0.93 0.42 -0.15 -2.21* -1.75 -0.10 

 Order 3.66 1.06 0.23 3.44*** 1.56 5.75 

 Global political orientation 6.19 2.01 0.21 3.08** 2.23 10.15 

*p < 0,05, **p < 0.01, ***p < 0.001

 Hence, partial correlations were conducted for each axis in order to control for covariance of the other axis when exploring the link between each axis and sensitivity to social norm violation. These analyses revealed a significant correlation between the identity axis and sensitivity to norm violation (r(197) = 0.15, p = 0.04), but not for the socioeconomic axis (r(197) = 0.08, p = 0.24).

 Since the identity axis was linked with sensitivity to social norm violation, we conducted a second hierarchical regression using identity scores instead of global political orientation scores. For this second regression, socioeconomic scores were added in the first step of the regression with the demographics variables previously mentioned. The general model was significant (F(5,193) = 6.43, p < 0.001) explaining 12.0% of the variance in sensitivity to social norm violation. The Identity axis contributed significantly to the model (ΔF(1,193) = 4.18, p = 0.042) and explained 1.9% of the variance. Hierarchical regression results are presented in Table 5.

Table 5. Coefficients of hierarchical regression analysis for variables predicting sensitivity to social norm violation with identity and socioeconomic axes (n=199)

 95% CI

Model Unstandardized β Standard Error Standardized β t Lower Upper

H₀ (Intercept) 29.31 8.73 3.37*** 12.09 46.54 

 Gender 1.73 0.69 0.17 2.51* 0.37 3.08 

 Education level -0.96 0.42 -0.15 -2.28* -1.79 -0.13 

 Order 3.81 1.07 0.24 3.57*** 1.71 5.92 

 Socioeconomic axis 4.46 1.54 0.20 2.90** 1.43 7.49 

H₁ (Intercept) 22.48 9.29 2.42* 4.17 40.79 

 Gender 1.78 0.68 0.18 2.61* 0.43 3.12 

 Education level -0.95 0.42 -0.15 -2.28* -1.78 -0.13 

 Order 3.70 1.06 0.23 3.49*** 1.61 5.79 

 Socioeconomic axis 3.15 1.65 0.14 1.91 -0.11 6.41 

 Identity axis 3.88 1.90 0.15 2.01* 0.14 7.61

---

## [Decision Letter · Decision Letter 1]

13 Nov 2020

Sensitivity to social norm violation is related to political orientation

PONE-D-20-22085R1

Dear Dr. Désilets,

We’re pleased to inform you that your manuscript has been judged scientifically suitable for publication and will be formally accepted for publication once it meets all outstanding technical requirements.

Thank you for your responsiveness to the reviewers and to my own concerns. I agree with R2 that the revision is nicely crafted.

Kind regards,

Rick K. Wilson, Ph.D.

Academic Editor

PLOS ONE

Additional Editor Comments (optional):

Reviewers' comments:

Reviewer's Responses to Questions

**Comments to the Author**

1. If the authors have adequately addressed your comments raised in a previous round of review and you feel that this manuscript is now acceptable for publication, you may indicate that here to bypass the “Comments to the Author” section, enter your conflict of interest statement in the “Confidential to Editor” section, and submit your "Accept" recommendation.

Reviewer #2: All comments have been addressed

2. Is the manuscript technically sound, and do the data support the conclusions?

Reviewer #2: Yes

3. Has the statistical analysis been performed appropriately and rigorously? 

Reviewer #2: Yes

4. Have the authors made all data underlying the findings in their manuscript fully available?

Reviewer #2: Yes

5. Is the manuscript presented in an intelligible fashion and written in standard English?

Reviewer #2: Yes

6. Review Comments to the Author

Reviewer #2: Review Comments to the Author: I appreciate all the work the authors did on this revision. I have no further comments on changes to be made. Great job!

7. PLOS authors have the option to publish the peer review history of their article (what does this mean?). If published, this will include your full peer review and any attached files.

Reviewer #2: No

---

## [Editor Report · Acceptance letter]

18 Nov 2020

PONE-D-20-22085R1 

Sensitivity to social norm violation is related to political orientation 

Dear Dr. Désilets:

I'm pleased to inform you that your manuscript has been deemed suitable for publication in PLOS ONE. Congratulations! Your manuscript is now with our production department. 

Kind regards, 

on behalf of

Dr. Rick K. Wilson 

Academic Editor

PLOS ONE